# DISCo: Distilled Student Models Co-training for Semi-supervised Text Mining

**Weifeng Jiang**[1,2,*]**, Qianren Mao**[2,†]**, Chenghua Lin**[3]**, Jianxin Li**[2,4]**, Ting Deng**[4]**, Weiyi Yang**[4]**, Zheng Wang**[5]

[1] SCSE, Nanyang Technological University, Singapore.

[2] Zhongguancun Laboratory, Beijing, P.R.China.

[3] Department of Computer Science, University of Manchester, U.K.

[4] School of Computer Science and Engineering, Beihang University, Beijing, P.R.China.

[5] The School of Computing, University of Leeds, U.K.

s220077@e.ntu.edu.sg, maoqr@zgclab.edu.cn

## Abstract

Many text mining models are constructed by fine-tuning a large deep pre-trained language model (PLM) in downstream tasks. However, a significant challenge nowadays is maintaining performance when we use a lightweight model with limited labelled samples. We present DISCo, a semi-supervised learning (SSL) framework for fine-tuning a cohort of small student models generated from a large PLM using knowledge distillation. Our key insight is to share complementary knowledge among distilled student cohorts to promote their SSL effectiveness. DISCo employs a novel co-training technique to optimize a cohort of multiple small student models by promoting knowledge sharing among students under diversified views: model views produced by different distillation strategies and data views produced by various input augmentations. We evaluate DISCo on both semi-supervised text classification and extractive summarization tasks. Experimental results show that DISCo can produce student models that are 7.6× smaller and 4.8× faster in inference than the baseline PLMs while maintaining comparable performance. We also show that DISCo-generated student models outperform the similar-sized models elaborately tuned in distinct tasks.

## 1 Introduction

Large pre-trained language models (PLMs), such as BERT (Devlin et al., 2019), GPT-3 (Brown et al., 2020), play a crucial role in the development of natural language processing applications, where one prominent training regime is to fine-tune the large and expensive PLMs for the downstream tasks of interest (Jiao et al., 2020).

Minimizing the model size and accelerating the model inference are desired for systems with limited computation resources, such as mobile (Liu et al., 2021) and edge (Tambe et al., 2021) devices. Therefore, maintaining the generalization ability of the reduced-sized model is crucial and feasible (Sun et al., 2019; Sanh et al., 2019; Jiao et al., 2020; Wang et al., 2020).

Semi-supervised learning (SSL) emerges as a practical paradigm to improve model generalization by leveraging both limited labelled data and extensive unlabeled data (Rasmus et al., 2015; Lee et al., 2013; Tarvainen and Valpola, 2017; Miyato et al., 2019; Berthelot et al., 2019; Sohn et al., 2020; Fan et al., 2023; Zhang et al., 2021; Berthelot et al., 2022; Zheng et al., 2022; Yang et al., 2023). While promising, combining SSL with a reduced-size model derived from PLMs still necessitates a well-defined learning strategy to achieve improved downstream performances (Wang et al., 2022a). This necessity arises because these shallow networks typically have lower capacity, and the scarcity of labeled data further curtails the model's optimization abilities. Besides, a major hurdle is a lack of labelled data samples – a particular problem for text mining tasks because the labelling text is labour-intensive and error-prone (Gururangan et al., 2019; Chen et al., 2020; Xie et al., 2020; Lee et al., 2021; Xu et al., 2022; Zhao et al., 2023).

This paper thus targets using SSL to leverage distilled PLMs in a situation where only limited labelled data is available and fast model inference is needed on resource-constrained devices. To this end, we use the well-established teacher-student knowledge distillation technique to construct small student models from a teacher PLM and then fine-tune them in the downstream SSL tasks. We aim to improve the effectiveness of fine-tuning small student models for text-mining tasks with limited labelled samples.

We present DISCo, a novel co-training approach aimed at enhancing the SSL performances by using distilled small models and few labelled data. The student models in the DISCo acquire complemen-

---

[*] Internship achievements at Zhongguancun Laboratory.

[†] Qianren Mao is the corresponding author.

tary information from multiple views, thereby improving the generalization ability despite the small model size and limited labelled samples. we introduce two types of view diversities for co-training: i) *model view diversity*, which leverages diversified initializations for student models in the cohort, ii) *data view diversity*, which incorporates varied noisy samples for student models in the cohort. Specifically, the model view diversity is generated by different task-agnostic knowledge distillations from the teacher model. The data view diversity is achieved through various embedding-based data augmentations to the input instances.

Intuitively, DɪsCo with the model view encourages the student models to learn from each other interactively and maintain reciprocal collaboration. The student cohort with the model views increases each participating model's posterior entropy (Chaudhari et al., 2017; Pereyra et al., 2017; Zhang et al., 2018), helping them to converge to a flatter minimum with better generalization. At the same time, DɪsCo with the data views regularizes student predictions to be invariant when applying noises to input examples. Doing so improves the models' robustness on diverse noisy samples generated from the same instance. This, in turn, helps the models to obtain missing inductive biases on learning behaviour, i.e., adding more inductive biases to the models can lessen their variance (Xie et al., 2020; Lovering et al., 2021).

We have implemented a working prototype of DɪsCo[1] and applied it to text classification and extractive summarization tasks. We show that by co-training just *two student models*, DɪsCo can deliver faster inference while maintaining the performance level of the large PLM. Specifically, DɪsCo can produce a student model that is 7.6× smaller (4-layer TinyBERT) with 4.8× faster inference time by achieving superior ROUGE performance in extractive summarization than the source teacher model (12-layer BERT). It also achieves a better or comparable text classification performance compared to the previous state-of-the-art (SOTA) SSL methods with 12-layer BERT while maintaining a lightweight architecture with only 6-layer Tiny-BERT. We also show that DɪsCo substantially outperforms other SSL baselines by delivering higher accuracy when using the same student models in model size.

## 2 Methodology

### 2.1 Overview of DɪsCo

DɪsCo jointly trains distilled student cohorts to improve model effectiveness in a complementary way from diversified views. As a working example, we explain how to use a dual-student DɪsCo to train two kinds of student models (see Figure 1). Extension to more students is straightforward (see section 2.3). To this end, DɪsCo introduces two initialization views during the co-training process: (*i*) model views which are different student model variants distilled from the teacher model, and (*ii*) data views which are different data augmented instances produced by the training input.

In DɪsCo, two kinds of compressed students (represented by two different colours in Figure 1(a)) are generated by the same teacher. This process allows us to pre-encode the model view specifically for DɪsCo. Additionally, we duplicate copies of a single student model to receive supervised and unsupervised data individually. In the supervised learning phase, DɪsCo optimizes two students using labelled samples. In the unsupervised learning phase, each student model concurrently shares the parameters with its corresponding duplicate, which is trained by supervised learning. The subsequent consistency training loss then optimizes the students using unlabeled samples.

For an ablation comparison of DɪsCo, we introduce the variant of DɪsCo only equipped with the model view, shown in Figure 1(b). In this variant, labelled and unlabeled data are duplicated and would be fed to the students directly. DɪsCo and its variant ensure reciprocal collaboration among the distilled students and can enhance the generalization ability of the student cohort by the consistency constraint. In this section, we introduce DɪsCo from two aspects: knowledge distillation and the co-training strategy.

### 2.2 Student Model Generation

Our current implementation uses knowledge distillation to generate small-sized models from a PLM. Like the task-agnostic distillation of Tiny-BERT[2] (Jiao et al., 2020), we use the original BERT without fine-tuning as the teacher model to generate the student models (In most cases, two student models at least are generated in our implementation). The task-agnostic distillation method is

[1]Code and data are available at: `https://github.com/LiteSSLHub/DisCo`.

[2]`https://github.com/huawei-noah/Pretrained-Language-Model/tree/master/TinyBERT`

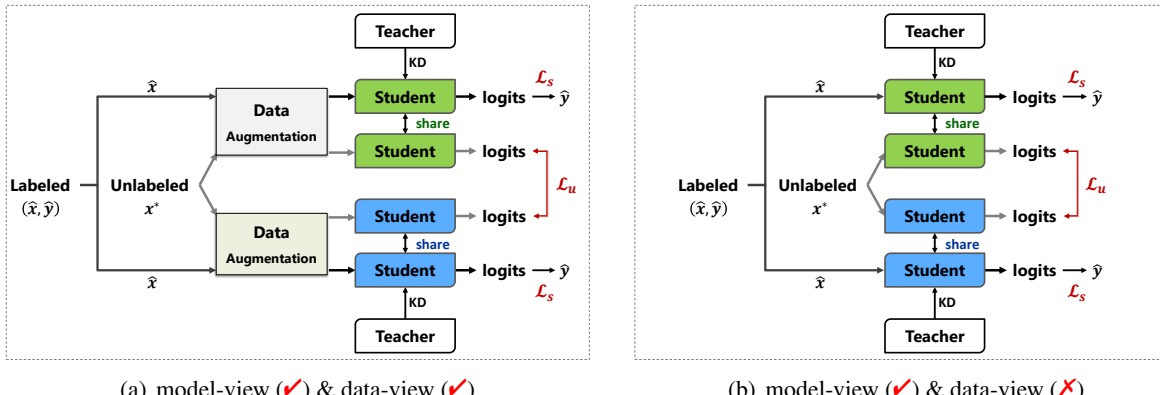

| (a) model-view (✔) & data-view (✔) | (b) model-view (✔) & data-view (✗) |

Figure 1: The training architecture of DISCO (a) and the ablation variant (b). ✔ refers to 'DO USE' the and ✗ is 'DO NOT USE' . $\mathcal{L}_s$ is a supervised loss and $\mathcal{L}_u$ is unsupervised. 'KD' is an abbreviation for knowledge distillation.

convenient for using any teacher network directly.

We use a large-scale general-domain corpus of `WikiText-103`[3] released by Merity et al. (2017) as the training data of the distillation. The student mimics the teacher's behaviour through the representation distillation from BERT layers: (*i*) the output of the embedding layer, (*ii*) the hidden states, and (*iii*) attention matrices.

### 2.2.1 Model View Encoding

To ensure the grouped students present a different view of the teacher, we distil different BERT layers from the same teacher. Model view encoding diversifies the individual student by leveraging different knowledge of the teacher. We propose two different strategies for the knowledge distillation process: (*i*) Separated-layer KD (SKD): the student learns from the alternate *k*-layer of the teacher. For instance, $\{3, 6, 9, 12\}$ are the 4 alternate layers of BERT. (*ii*) Connected-layer KD (CKD): the student learns from the continuous *K*-layer of the teacher. For example, $\{1, 2, 3, 4\}$ are the continuous 4 layers of BERT. In the case of dual-student DISCO, the two students with two kinds of knowledge distillation strategies are represented as $S^{AK}$ and $S^{BK}$. The co-training framework will encourage the distinct individual model to teach each other in a complementary manner underlying model view initialization.

With consistency constraints, our co-training framework can obtain valid inductive biases on model views, enabling student peers to teach each other and to generalize unseen data. Apart from the model views, we also introduce data views produced by various data augmentations of inputs to expand the inductive biases.

### 2.2.2 Data View Encoding

We use different data augmentation strategies at the token embedding layer to create different data views from the input samples. Our intuition is that advanced data augmentation can introduce extra inductive biases since they are based on random sampling at the token embedding layer with minimal semantic impact (Xie et al., 2020; Wu et al., 2020; Yan et al., 2021; Gao et al., 2021). Inspired by ConSERT (Yan et al., 2021) and Sim-CSE (Gao et al., 2021), we adopt convenient data augmentation methods: adversarial attack (Kurakin et al., 2017), token shuffling (Lee et al., 2020), cutoff (Shen et al., 2020) and dropout (Hinton et al., 2012), described as follows.

**Adversarial Attack (`AD`).** We implement it with Smoothness-Inducing Adversarial Regularization (SIAR)[4] (Jiang et al., 2020), which encourages the model's output not to change too much when a small perturbation is injected to the input.

**Token Shuffling (`TS`).** This strategy is slightly similar to Lee et al. (2020) and Yan et al. (2021), and we implement it by passing the shuffled position IDs to the embedding layer while keeping the order of the token IDs unchanged.

**Cutoff (`CO`).** This method randomly erases some tokens for token cutoff in the embedding matrix.

**Dropout (`DO`).** As same as in BERT, this scheme randomly drops elements by a specific probability and sets their values to zero.

DISCO incorporates two forms of data view during co-training: a `HARD FORM` and a `SOFT FORM`. Taking dual-student networks for example, we use

---

[3]https://huggingface.co/datasets/wikitext

[4]The adversarial perturbed embeddings are generated in the AD strategy by maximizing the supervised loss.

| Datasets | Label Type | #Classes | #Labeled | #Unlabelled | #Dev | #Test |
|---|---|---|---|---|---|---|
| CNN/DailyMail | Extractive Sentences | 3 | 10/100/1,000 | 287,227 -10/-100/-1,000 | 13,368 | 11,490 |
| Agnews | News Topic | 4 | $\times 10/ \times 30/ \times 100$ | 20,000 | 8,000 | 7,600 |
| Yahoo!Answer | Q&A Topic | 10 | $\times 10/ \times 30/ \times 100$ | 50,000 | 20,000 | 59,727 |
| DBpedia | Wikipedia Topic | 14 | $\times 10/ \times 30/ \times 100$ | 70,000 | 28,000 | 70,000 |

Table 1: Dataset statistics and dataset split of the semi-supervised extractive summarization dataset and several typical semi-supervised text classification datasets, in which '$\times$' means the number of data per class.

two different data augmentation approaches, such as AD and DO, to implement the HARD FORM data view. Regarding the SOFT FORM data view, we apply the same data augmentation approach, including AD with two rounds of random initialization to ensure distinct views. In DisCo, each student obtains perturbation differences through the various combinations of the HARD FORM and SOFT FORM.

### 2.2.3 Co-training Framework

Formally, we are provided with a semi-supervised dataset $\mathcal{D}$, $\mathcal{D} = \mathcal{S} \cup \mathcal{U}$. $\mathcal{S} = \{(\hat{x}, \hat{y})\}$ is labelled data, where $(\hat{x}, \hat{y})$ will be used for two kinds of students identically. $\mathcal{U} = \{x^*\}$ is unlabeled data, and two copies are made for two kinds of students identically. For $X \in \mathcal{D}$, let $\phi_A(X)$ and $\phi_B(X)$ denote the two data views of data $X$. A pair of models ($S^{AK} = f_A$ and $S^{BK} = f_B$) are two distilled student models which we treat as the model view of dual-student DisCo. Student $f_A$ only uses $\phi_A(X)$, and Student $f_B$ uses $\phi_B(X)$.

By training collaboratively with the cohort of students $f_A$ and $f_B$, the co-training optimization objective allows them to share the complementary information, which improves the generalization ability of a network.

**Supervised Student Cohort Optimization**. For supervised parts, we use the categorical Cross-Entropy (CE) loss function for optimizing student $f_A$ and student $f_B$, respectively. They are trained with the labeled data $(\hat{x}, \hat{y})$ sampled from $\mathcal{S}$.

$$\mathcal{L}_{s_A} = \text{CE}(f_A(\phi_A(\hat{x})), \hat{y}), \quad (1)$$

$$\mathcal{L}_{s_B} = \text{CE}(f_B(\phi_B(\hat{x})), \hat{y}). \quad (2)$$

**Unsupervised Student Cohort Optimization**. In standard co-training, multiple classifiers are expected to provide consistent predictions on unlabeled data $x^* \in \mathcal{U}$.

The consistency cost of the unlabeled data $x^*$ is computed from the two student output logits: $z_A(\phi_A(x^*))$ and $z_B(\phi_B(x^*))$. We use the Mean

Square Error (MSE) to encourage the two students to predict similarly:

$$\mathcal{L}_{u_{A,B}} = \text{MSE}(z_A(\phi_A(x^*)), z_B(\phi_B(x^*))), \quad (3)$$

$$\mathcal{L}_{u_{B,A}} = \text{MSE}(z_B(\phi_B(x^*)), z_A(\phi_A(x^*))). \quad (4)$$

**Overall Training Objective**. Finally, we combine supervised cross-entropy loss with unsupervised consistency loss and train the model by minimizing the joint loss:

$$\mathcal{L}_{\Theta} = \mathcal{L}_{s_A} + \mathcal{L}_{s_B} + \mu(t, n) \cdot \lambda \cdot (\mathcal{L}_{u_{A,B}} + \mathcal{L}_{u_{B,A}}), \quad (5)$$

where $\mu(t, n) = min(\frac{t}{n}, 1)$. It represents the ramp-up weight starting from zero, gradually increasing along with a linear curve during the initial $n$ training steps. $\lambda$ is the hyperparameter balancing supervised and unsupervised learning.

### 2.3 Co-training of Multi-student Peers

So far, our discussion has been focused on training two students. DisCo can be naturally extended to support not only two students in the student cohort but more student networks. Given $K$ networks $\Theta_1, \Theta_2, ..., \Theta_K(K \geq 2)$, the objective function for optimising all $\Theta_k$, $(1 \leq k \leq K)$, becomes:

$$\mathcal{L}_{\Theta} = \sum_{k=1}^{K} \left( \mathcal{L}_{s_k} + \mu(t, n) \cdot \lambda \cdot \mathcal{L}_{u_{i,k}} \right), \quad (6)$$

$$\mathcal{L}_{u_{i,k}} = \frac{1}{K-1} \sum_{i=1, i \neq k}^{K} \text{MSE}(z_i(\phi_i(x^*)), z_k(\phi_k(x^*))). \quad (7)$$

Equation (5), is now a particular case of (6) with $k = 2$. With more than two networks in the cohort, a learning strategy for each student of DisCo takes the ensemble of other $K-1$ student peers to provide mimicry targets. Namely, each student learns from all other students in the cohort individually.

| Models | $\mathcal{P}$ | Agnews | | | Yahoo!Answer | | | DBpedia | | | Avg |
|---|---|---|---|---|---|---|---|---|---|---|---|
| | | 10 | 30 | 200 | 10 | 30 | 200 | 10 | 30 | 200 | |
| $\text{BERT}_{\text{BASE}}$ | 109.48M | 81.00 | 84.32 | 87.24 | 60.10 | 64.13 | 69.28 | 96.59 | 98.21 | 98.79 | 82.18 |
| UDA | 109.48M | 84.70 | 86.89 | 88.56 | 64.28 | 67.70 | 69.71 | 98.13 | 98.67 | 98.85 | 84.17 |
| $\text{TinyBERT}^6$ | 66.96M | 71.45 | 82.46 | 87.59 | 52.84 | 60.59 | 68.71 | 96.89 | 98.16 | 98.65 | 79.70 |
| $\text{UDA}_{\text{TinyBERT}^6}$ | 66.96M | 73.90 | 85.16 | 87.54 | 57.14 | 62.86 | 67.93 | 97.41 | 97.87 | 98.26 | 81.79 |
| DɪsCo ($\text{S}^{\text{A6}}$) | 66.96M | 74.38 | 86.39 | 88.70 | 57.62 | 64.04 | 69.57 | 98.50 | 98.45 | 98.57 | 82.02 |
| DɪsCo ($\text{S}^{\text{B6}}$) | 66.96M | **77.45** | **86.93** | **88.82** | **59.10** | **66.58** | **69.75** | **98.57** | **98.61** | **98.73** | **82.73** |
| $\text{TinyBERT}^4$ | 14.35M | 69.67 | 78.35 | 85.12 | 42.66 | 53.63 | 61.89 | 89.65 | 96.88 | 97.58 | 75.05 |
| $\text{UDA}_{\text{TinyBERT}^4}$ | 14.35M | 69.60 | 77.56 | 83.60 | 40.69 | 55.43 | 63.34 | 88.50 | 93.63 | 95.98 | 74.26 |
| DɪsCo ($\text{S}^{\text{A4}}$) | 14.35M | 76.90 | 85.39 | 87.82 | **51.48** | 62.36 | 68.10 | 94.02 | 98.13 | 98.56 | 80.31 |
| DɪsCo ($\text{S}^{\text{B4}}$) | 14.35M | **77.36** | **85.55** | **87.95** | 51.31 | **62.93** | **68.24** | **94.79** | **98.14** | **98.63** | **80.54** |
| FLiText | 9.60M | 67.14 | 77.12 | 82.12 | 48.30 | 57.01 | 63.09 | 89.26 | 94.04 | 97.01 | 75.01 |
| DɪsCo ($\text{S}^{\text{A2}}$) | 8.90M | 70.61 | 81.87 | 86.08 | 48.41 | 57.84 | 64.04 | **89.67** | 96.06 | 97.58 | 76.90 |
| DɪsCo ($\text{S}^{\text{B2}}$) | 8.90M | **75.05** | **82.16** | **86.38** | **51.05** | **58.83** | **65.63** | 89.55 | **96.14** | **97.70** | **78.05** |

Table 2: Text classification performance (Acc (%)) on typical semi-supervised text classification tasks. $\mathcal{P}$ is the number of model parameters. The best results are in-bold.

## 3 Experiments

**Datasets.** We evaluate DɪsCo on extractive summarization and text classification tasks, as shown in Table 1. For extractive summarization, we use the CNN/DailyMail (Hermann et al., 2015) dataset, training the model with 10/100/1000 labeled examples. Regarding text classification, we evaluate on semi-supervised datasets: Agnews (Zhang et al., 2015) for News Topic classification, Yahoo!Answers (Chang et al., 2008) for Q&A topic classification, and DBpedia (Mendes et al., 2012) for WikiPedia topic classification. The models are trained with 10/30/200 labeled data per class and 5000 unlabeled data per class. Further details on the evaluation methodology are in Appendix A.3.

**Implementation Details.** The main experimental results presented in this paper come from the best model view and data view we found among multiple combinations of view encoding strategies. Taking dual-students DɪsCo as an example, we present the results of $\text{S}^{\text{A}K}$ and $\text{S}^{\text{B}K}$, as the model-view being a combination of SKD (alternate $K$-layer) and CKD (continuous $K$-layer). The data view is the SOFT FORM of two different AD initialization. Specifically , DɪsCo ($\text{S}^{\text{A6}}$) uses CKD model-view of BERT layers $\{1, 2, 3, 4, 5, 6\}$ and SOFT FORM data-view of AD. DɪsCo ($\text{S}^{\text{B6}}$) uses SKD model-view of BERT layers $\{2, 4, 6, 8, 10, 12\}$ and SOFT FORM data-view of AD. DɪsCo ($\text{S}^{\text{A4}}$ and $\text{S}^{\text{B4}}$) use similar combinations to the DɪsCo ($\text{S}^{\text{A6}}$ and $\text{S}^{\text{B6}}$). DɪsCo ($\text{S}^{\text{A2}}$) uses CKD with $\{1, 2\}$ BERT layers and DɪsCo ($\text{S}^{\text{B2}}$) uses CKD with $\{11, 12\}$ BERT layers. The details of DɪsCo's hyperparameter are presented in Appendix A.2. We run each experiment with three random seeds and report the mean performance on test data and the experiments are conducted on a single NVIDIA Tesla V100 32GB GPU.

**Competing Baselines.** For text classification tasks, we compare DɪsCo with: (*i*) supervised baselines, $\text{BERT}_{\text{BASE}}$ and default TinyBERT (Jiao et al., 2020), (*ii*) semi-supervised UDA (Xie et al., 2020) and FLiText (Liu et al., 2021). We also compare with other prominent SSL text classification methods and report their results on the Unified SSL Benchmark (USB) (Wang et al., 2022a). Most of these SSL methods work well on computer vision (CV) tasks, and Wang et al. (2022a) generalize them to NLP tasks by integrating a 12-layer BERT. More detailed introductions are given in Appendix A.4. For extractive summarization tasks, we compare: (*i*) supervised basline, BERTSUM (Liu and Lapata, 2019), (*ii*) two SOTA semi-supervised extractive summarization methods, UDASUM and CPSUM (Wang et al., 2022b), (*iii*) three unsupervised techniques, LEAD-3, TextRank (Mihalcea and Tarau, 2004) and LexRank (Erkan and Radev, 2004). We use the open-source releases of the competing baselines.

## 4 Experimental Results

### 4.1 Evaluation on Text Classification

As shown in Table 2, the two students produced by DɪsCo with a 6-layer distilled BERT ($\text{S}^{\text{A6}}$ and $\text{S}^{\text{B6}}$) consistently outperform TinyBERT and $\text{UDA}_{\text{TinyBERT}}$ in all text classification tasks. Moreover, one student of our dual-student 6-layer DɪsCo

Table 3: Text classification performance (Acc (%)) of other prominent SSL text classification models and all results reported by the Unified SSL Benchmark (USB) (Wang et al., 2022a). $\mathcal{D}$ refers to datasets, $L_m$ is the number of the BERT layers used by models and $L_d$ is labeled data per class.

| $\mathcal{D}$ | Models | $\mathbf{L}_m$ | $\mathbf{L}_d$ | Acc |
|---|---|---|---|---|
| Agnews | ∏-model (Rasmus et al., 2015) | | 50 | 86.56 |
| | P-Labeling (Lee et al., 2013) | | 50 | 87.01 |
| | MeanTeacher (Tarvainen and Valpola, 2017) | 12 | 50 | 86.77 |
| | PCM (Xu et al., 2022) | | 30 | **88.42** |
| | MixText (Chen et al., 2020) | | 30 | 87.40 |
| | DɪsCo (ours) | 6 | 30 | **86.93** |
| Yahoo!Answer | AdaMatch (Berthelot et al., 2022) | | 200 | 69.18 |
| | CRMactch (Fan et al., 2023) | | 200 | 69.38 |
| | SimMatch (Zheng et al., 2022) | | 200 | 69.36 |
| | FlexMatch (Zhang et al., 2021) | 12 | 200 | 68.58 |
| | VAT (Miyato et al., 2019) | | 200 | 68.47 |
| | MeanTeacher (Tarvainen and Valpola, 2017) | | 200 | 66.57 |
| | DɪsCo (ours) | 6 | 200 | **69.75** |
| DBpedia | PCM (Xu et al., 2022) | | 10 | **98.70** |
| | Mixtext (Chen et al., 2020) | 12 | 10 | 98.39 |
| | VAT (Miyato et al., 2019) | | 10 | 98.40 |
| | DɪsCo (ours) | 6 | 10 | **98.57** |

outperforms the 12-layer supervised $BERT_{BASE}$ by a 0.55% average improvement in accuracy. These results suggest that DɪsCo provides a simple but effective way to improve the generalization ability of small networks by training collaboratively with a cohort of other networks.

Compared with the FLiText, DɪsCo improves the average classification accuracy by 1.9% while using a student model with 0.7M fewer parameters than FLiText. FLiText relies heavily on back-translation models for generating augmented data, similar to UDA. Unfortunately, this strategy fails to eliminate error propagation introduced by the back-translation model and requires additional data pre-processing. Besides, FLiText consists of two training stages and needs supervised optimization in both stages, increasing training costs and external supervised settings.

Table 3 shows results when comparing DɪsCo to other prominent SSL methods which are integrated with a 12-layer BERT. We take the results from the source publication or Unified SSL Benchmark (USB) (Wang et al., 2022a) for these baselines. However, most of them perform worse than DɪsCo's students only with a 6-layer BERT using same labeled data. In the case of Yahoo!Answer text classification, our 6-layer BERT-

Table 4: ROUGE F1 performance of the extractive summarization. $L_d$=100 refers to the labeled data per class. SSL baselines (CPSUM and UDASUM) use the same unlabeled data as DɪsCo has used.

| Models | $\mathcal{P}$ | $\mathbf{L}_d$ | CNN/DailyMail R-1 | R-2 | R-L |
|---|---|---|---|---|---|
| ORACLE | | 100 | *48.35* | *26.28* | *44.61* |
| LEAD-3 | | 100 | 40.04 | 17.21 | 36.14 |
| TextRank | | 100 | 33.84 | 13.11 | 23.98 |
| LexRank | | 100 | 34.63 | 12.72 | 21.25 |
| BERTSUM | 109.48M | 100 | 38.58 | 15.97 | 34.79 |
| CPSUM | 109.48M | 100 | 38.10 | 15.90 | 34.39 |
| UDASUM | 109.48M | 100 | 38.58 | 15.87 | 34.78 |
| $TinyBERTSUM^4$ | 14.35M | 100 | 39.83 | 17.24 | 35.98 |
| $TinyBERTSUM^{F4}$ | 14.35M | 100 | 40.06 | 17.32 | 36.18 |
| $TinyBERTSUM^{L4}$ | 14.35M | 100 | 39.88 | 17.14 | 36.00 |
| $UDASUM_{TinyBERT^4}$ | 14.35M | 100 | 40.11 | 17.43 | 36.23 |
| $UDASUM_{TinyBERT^{A4}}$ | 14.35M | 100 | 39.90 | 17.25 | 36.05 |
| $UDASUM_{TinyBERT^{B4}}$ | 14.35M | 100 | 40.11 | 17.34 | 36.19 |
| DɪsCo ($S^{A4}$) | 14.35M | 100 | 40.39 | 17.57 | 36.47 |
| DɪsCo ($S^{B4}$) | 14.35M | 100 | **40.41** | **17.59** | **36.50** |

Table 5: Model efficiency about the model size and inference speedup on a single NVIDIA Tesla V100 32GB GPU. $\mathcal{T}_{TS}$(ms) refers to the speedup of extractive summarization models trained with 100 labeled data. $\mathcal{T}_{TC}$(ms) illustrates the speedup of text classification models trained with Agnews 200 labeled data per class.

| Models | $\mathcal{T}_{TS}$(ms) | Models | $\mathcal{T}_{TC}$(ms) |
|---|---|---|---|
| BERTSUM | 12.66 | $BERT_{BASE}$ | 12.94 |
| CPSUM | 12.66 | $TinyBERT^4$ | 2.86 |
| $TinyBERTSUM^4$ | 2.64 | $UDA_{TinyBERT^4}$ | 2.86 |
| $UDASUM_{TinyBERT^4}$ | 2.64 | FLiText | 1.56 |
| DɪsCo ($S^{A4}$ or $S^{B4}$) | 2.64 | DɪsCo ($S^{A2}$ or $S^{B2}$) | 1.72 |

based DɪsCo achieves better performance than all 12-layer BERT-based SSL benchmarks. These results demonstrate that our model has superiority in certain scenarios of the lightweight model architecture and limited manual annotation.

## 4.2 Evaluation on Extractive Summarization

For the semi-supervised extractive summarization tasks, our dual-student DɪsCo outperforms all baselines in Table 4. Despite using a smaller-sized, 4-layer model, DɪsCo performs better than the 12-layer BERTSUM, UDA, and CPSUM. The results show that our methods can reduce the cost of supervision in extractive summarization tasks. Other ROUGE results with 10 or 1000 labeled examples are presented in Appendix A.5.

Table 6: Text classification performance (Acc (%)) of DɪsCo with multiple student peers. The students (S$^{A2}$, S$^{B2}$, S$^{C2}$, S$^{D2}$) are distilled from layers {1, 2}, {3, 4}, {9, 10} and {11, 12} of the teacher BERT$_{BASE}$, respectively. The first four students adopt HARD FORM data views which are AD, DO, TS, and CO, respectively. The last four students adopt a SOFT FORM data view with different DO initialization. Better results than dual-student DɪsCo in Table 2 is in-bold.

| Models | $L_d$ | Agnews | Yahoo!Answer | DBpedia |
|---|---|---|---|---|
| DɪsCo (S$^{A2}$) | 200 | **87.58** | **66.74** | **98.23** |
| DɪsCo (S$^{B2}$) | 200 | **87.41** | **66.28** | **98.33** |
| DɪsCo (S$^{C2}$) | 200 | **87.83** | 65.63 | 97.69 |
| DɪsCo (S$^{D2}$) | 200 | **87.59** | **65.87** | **98.34** |
| DɪsCo (S$^{A2}$) | 200 | **86.99** | **65.71** | **98.10** |
| DɪsCo (S$^{B2}$) | 200 | **86.71** | 64.01 | **98.18** |
| DɪsCo (S$^{C2}$) | 200 | **86.79** | 63.96 | **98.12** |
| DɪsCo (S$^{D2}$) | 200 | **86.63** | 63.83 | **98.01** |

Table 7: Performance comparison between DɪsCo and a single student model with AD augmentation. The 'SingleStudent' is the better-performing model among the two students within the DɪsCo framework.

| Models | $L_d$ | Agnews | Yahoo!Answer | DBpedia |
|---|---|---|---|---|
| SingleStudent[6] | 10 | 73.52 | 55.43 | 93.65 |
| DɪsCo (S$^{A6}$) | 10 | 74.38 | 57.62 | 98.50 |
| DɪsCo (S$^{B6}$) | 10 | **77.45** | **59.10** | **98.57** |
| SingleStudent[4] | 10 | 75.49 | 47.57 | 89.30 |
| DɪsCo (S$^{A4}$) | 10 | 76.90 | **51.48** | 94.02 |
| DɪsCo (S$^{B4}$) | 10 | **77.36** | 51.31 | **94.79** |
| SingleStudent[2] | 10 | 68.79 | 48.87 | 77.26 |
| DɪsCo (S$^{A2}$) | 10 | 70.61 | 48.41 | **89.67** |
| DɪsCo (S$^{B2}$) | 10 | **75.05** | **51.05** | 89.55 |

(multi-students) in the co-training process yields a considerable performance enhancement compared to a less populous student group (dual-students).

### 4.3 Model Efficiency

As shown in Table 5, compared with the teacher BERT$_{BASE}$, all 4-layer student models give faster inference time by speeding up the inference by 4.80×-7.52× for the two tasks. FLiText is slightly faster than the smaller model generated DɪsCo. This is because FLiText uses a convolutional network while our student models use BERT with multi-head self-attention. The lower computational complexity of convolutional networks[5]. However, despite the FLiText having more parameters, it gives worse performance (about 3.04% accuracy defects on average), as shown in Table 2.

### 4.4 Ablation Studies

#### 4.4.1 Effect of using Multi-student Peers

Having examined the dual-student DɪsCo in prior experiments, our next focus is to explore the scalability of DɪsCo by introducing more students in the cohort. As the results are shown in Table 6, we can see that the performance of every single student improves with an extension to four students in the DɪsCo cohort, which demonstrates that the generalization ability of students is enhanced when they learn together with increasing numbers of peers.

Besides, the results in Table 6 have validated the necessity of co-training with multiple students. It is evident that a greater number of student peers

#### 4.4.2 Effect of using Multi-View Strategy

As shown in Table 8, DɪsCo composed of the student networks distilled from the teacher is obviously superior to DɪsCo composed of two randomly initialized student networks, which verifies the advantage of our model view settings. In DɪsCo, the data view of SOFT FORM and HARD FORM brings the best effect, namely combinations of DO and AD encoded data view. Other data views with combinations of TS and CO yielded sub-optimal effects, which are presented in Appendix A.5. Under the same model view, DɪsCo integrating with the SOFT FORM data view is slightly better than the one using HARD FORM data view. The observations indicate adversarial perturbations are more useful for dual-student DɪsCo. Modelling the invariance of the internal noise in the sentences can thus improve the model's robustness.

Further, we plot the training loss contour of DɪsCo and its ablation model in Figure 2. Both models have a fair benign landscape dominated by a region with convex contours in the centre and no dramatic non-convexity. We observe that the optima obtained by training with the model view and the data view are flatter than those obtained only with a model view. A flat landscape implies that the small perturbations of the model parameters cannot hurt the final performance seriously, while a chaotic landscape is more sensitive to subtle changes (Li et al., 2018).

---

[5] The 1D-CNN requires $O(k \times n \times d)$ operations used by FLiText. In contrast, the multi-head self-attention mechanism of BERT requires $O(n^2 \times d + n \times d^2)$ operations, where $n$ is the sequence length, $d$ is the representation dimension, $k$ is the kernel size of convolutions.

Table 8: The impact of incorporating multi-view encoding for the dual-student DısCo. The `HARD` data-view is created using dropout (`DO`) and adversarial attack (`AD`). The `SOFT` view employs adversarial attack (`AD`) with varying initialization. The model-view (✔) refers to that students are trained from scratch without any teacher knowledge.

| model view | | data view | | CNN/DailyMail w. 100 | | | | | | Agnews w. 10 | | Yahoo!Answer w. 10 | |
|---|---|---|---|---|---|---|---|---|---|---|---|---|---|
| | | | | $S^{A4}$, R-1 | $S^{B4}$, R-1 | $S^{A4}$, R-2 | $S^{B4}$, R-2 | $S^{A4}$, R-L | $S^{B4}$, R-L | $S^{A4}$, ACC | $S^{B4}$, ACC | $S^{A4}$, ACC | $S^{B4}$, ACC |
| ✔ | | ✗ | | 36.74 | 36.69 | 14.15 | 14.12 | 32.91 | 32.86 | 37.51 | 36.76 | 22.23 | 21.62 |
| ✔ | $S^{A4}/S^{B4}$ | ✗ | | 39.96 | 39.93 | 17.23 | 17.23 | 36.07 | 36.06 | 73.18 | 73.62 | 52.56 | 52.95 |
| ✗ | $S^{A4}/S^{A4}$ | ✔ | HARD | 40.06 | 40.09 | 17.30 | 17.33 | 36.18 | 36.19 | 74.06 | 73.51 | 51.44 | 50.16 |
| ✗ | $S^{B4}/S^{B4}$ | ✔ | HARD | 40.16 | 40.17 | 17.35 | 17.36 | 36.26 | 36.26 | 77.45 | 77.22 | 54.02 | 53.80 |
| ✔ | $S^{A4}/S^{B4}$ | ✔ | HARD | 40.28 | 40.24 | 17.46 | 17.46 | 36.37 | 36.33 | 77.45 | 77.77 | 56.22 | 55.43 |
| ✗ | $S^{A4}/S^{A4}$ | ✔ | SOFT | 40.16 | 40.13 | 17.39 | 17.37 | 36.26 | 36.23 | 77.28 | 76.70 | 51.66 | 51.77 |
| ✗ | $S^{B4}/S^{B4}$ | ✔ | SOFT | 40.28 | 40.27 | 17.46 | 17.45 | 36.36 | 36.35 | 76.99 | 77.03 | 55.35 | 55.59 |
| ✔ | $S^{A4}/S^{B4}$ | ✔ | SOFT | 40.32 | 40.31 | 17.52 | 17.52 | 36.41 | 36.40 | 77.18 | 77.45 | 55.76 | 55.44 |

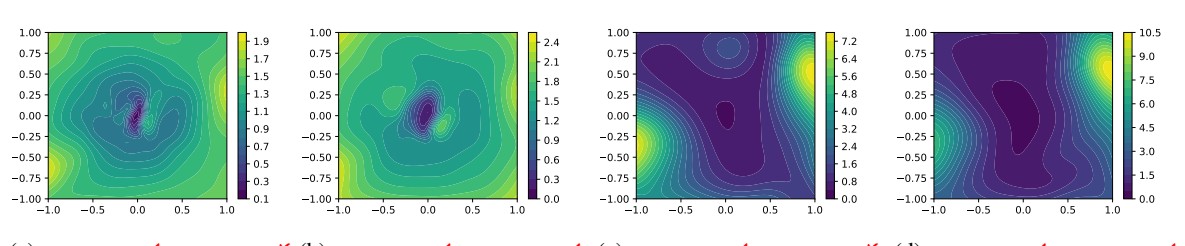

(a) model-view (✔) & data-view (✗) (b) model-view (✔) & data-view (✔) (c) model-view (✔) & data-view (✗) (d) model-view (✔) & data-view (✔)

Figure 2: 2D visualization of the loss surface contour of DısCo (w. model view and w. data view) and its ablation variant (w. model view). Subfigures (a) and (b) are the text classification tasks for Agnews dataset with 10 labeled data per class. Subfigures (c) and (d) are the extractive summarization tasks with 100 labeled data.

Table 9: Performance comparison (Acc (%)) of the back-translation (BT) and Adversarial Attack (`AD`) augmentation methods within the UDA and FLiText frameworks.

| Models | Aug | $L_d$ | Agnews | Yahoo!Answer | DBpeida |
|---|---|---|---|---|---|
| UDA$_{TinyBERT^6}$ | BT | 10 | 73.90 | 57.14 | 97.41 |
| | AD | 10 | 61.20 | 52.29 | 88.76 |
| FLiText | BT | 10 | 67.14 | 48.30 | 89.26 |
| | AD | 10 | 65.15 | 48.06 | 85.17 |

### 4.5 Discussion

#### 4.5.1 Single Student with `AD` Augmentation

To demonstrate the necessity of multi-student co-training, we compare the single-student model without co-training with `AD` data augmentations. Naturally, the single model exclusively uses supervised data, missing out on leveraging unsupervised data. A noteworthy performance decline is observed in Table 7 and most differently sized models in DBpedia suffer noticeable performance drops. These results validate the DısCo framework's efficacy under co-training optimization.

#### 4.5.2 UDA/FLiText with `AD` Augmentation

In the preceding analysis detailed in Table 2, UDA/FLiText utilized back translation as their data augmentation strategy, a technique distinctly different from the token embedding level data augmentation employed in our DısCo framework. To ensure a balanced comparison, we substituted the back translation approach with our `AD` augmentation method for UDA/FLiText. The outcomes of this modification are portrayed in Table 9.

These results underscore that regardless of the data augmentation strategy implemented, the performance of both UDA and FLiText falls short compared to our DısCo framework. This substantiates our claim that our co-training framework is superior in distilling knowledge encapsulated in unsupervised data. Furthermore, the performance across most tasks experiences a decline after the augmentation technique alteration. As stipulated in (Xie et al., 2020), the UDA/FLiText framework necessitates that augmented data maintain 'similar semantic meanings' thereby making back-translation a more suitable for UDA/FLiText, compared to the `AD` augmentation we incorporated.

### 5 Conclusion

In this paper, we present DısCo, a framework of co-training distilled students with limited labelled data, which is used for targeting the lightweight

models for semi-supervised text mining. DisCo leverages model views and data views to improve the model's effectiveness. We evaluate DisCo by applying it to text classification and extractive summarization tasks and comparing it with a diverse set of baselines. Experimental results show that DisCo substantially achieves better performance across scenarios using lightweight SSL models.

## 6 Limitations

Naturally, there is room for further work and improvement, and we discuss a few points here. In this paper, we apply DisCo to BERT-based student models created from the BERT-based teacher model. It would be useful to evaluate if our approach can generalize to other model architectures like TextCNN (Kim, 2014) and MLP-Mixer (Tolstikhin et al., 2021). It would also be interesting to extend our work to utilize the inherent knowledge of other language models (e.g., RoBERTa (Liu et al., 2019), GPT (Radford et al., 2018; Radford et al.; Brown et al., 2020), T5 (Raffel et al., 2020)).

Another limitation of our framework settings is the uniform number of BERT layers in all distilled student models. To address this, students in DisCo can be enhanced by introducing architectural diversity, such as varying the number of layers. Previous studies (Mirzadeh et al., 2020; Son et al., 2021) have demonstrated that a larger-size student, acting as an assistant network, can effectively simulate the teacher and narrow the gap between the student and the teacher. We acknowledge these limitations and plan to address them in future work.

## 7 Ethical Statement

The authors declare that we have no conflicts of interest. Informed consent is obtained from all individual participants involved in the study. This article does not contain any studies involving human participants performed by any authors.

## Acknowledgements

This work is supported in part by the National Natural Science Foundation of China (No.U20B2053).

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

# A Appendix

## A.1 Background and Related Work

**Knowledge Distillation (KD).** The KD (Hinton et al., 2015) is one of the promising ways to transfer from a powerful large network or ensemble to a small network to meet the low-memory or fast execution requirements. BANs (Furlanello et al., 2018) sequentially distill the teacher model into multiple generations of student models with identical architecture to achieve better performance. BERT-PKD (Sun et al., 2019) distills patiently from multiple intermediate layers of the teacher model at the fine-tuning stage. DistilBERT (Sanh et al., 2019) and MiniLM (Wang et al., 2020) leverage knowledge distillation during the pre-training stage. TinyBERT (Jiao et al., 2020) sets a two-stage knowledge distillation procedure that contains general-domain and tasks-specific distillation in Transformer (Vaswani et al., 2017). Despite their success, they may encounter difficulties affecting the sub-optimal performance in language understanding tasks due to the trade-off between model compression and performance loss.

**Semi-supervised Learning (SSL).** The majority of SSL algorithms are primarily concentrated in the field of computer vision, including Pseudo Labeling (Lee et al., 2013), Mean Teacher (Tarvainen and Valpola, 2017), VAT (Miyato et al., 2019), Mix-Match (Berthelot et al., 2019), FixMatch (Sohn et al., 2020), CRMatch (Fan et al., 2023), Flex-Match (Zhang et al., 2021), AdaMatch (Berthelot et al., 2022), and SimMatch (Zheng et al., 2022), all of which exploit unlabeled data by encouraging invariant predictions to input perturbations (Sajjadi et al., 2016). The success of semi-supervised learning methods in the visual area motivates research in the NLP community. Typical techniques include VAMPIRE (Gururangan et al., 2019), Mix-Text (Chen et al., 2020) and UDA (Xie et al., 2020). Under the low-density separation assumption, these SSL methods perform better than their fully-supervised counterparts while using only a fraction of labelled samples.

**Co-Training.** It is a classic award-winning method for semi-supervised learning paradigm, training two (or more) deep neural networks on complementary views (i.e., data view from different sources that describe the same instances) (Blum and Mitchell, 1998). By minimizing the error on limited labelled examples and maximizing the agreement on sufficient unlabeled examples, the co-training framework finally achieves two accurate classifiers on each view in a semi-supervised manner (Qiao et al., 2018).

## A.2 Hyperparameters

The $BERT_{BASE}$, as the teacher model, has a total of 109M parameters (the number of layers $N = 12$, the hidden size $d = 768$, the forward size $d' = 3072$ and the head number $h = 12$). We used the BERT tokenizer[6] to tokenize the text. The source text's max sentence length is 512 for extractive summarization and 256 for text classification. For extractive summarization, we select the top 3 sentences according to the average length of the Oracle human-written summaries. We use the default dropout settings in our distilled BERT architecture. The ratio of token cutoff is set to 0.2, as suggested in (Yan et al., 2021; Shen et al., 2020). The ratio of dropout is set to 0.1. Adam optimizer with $\beta_1 = 0.9, \beta_2 = 0.999$ is used for fine-tuning. We set the learning rate 1e-4 for extractive summarization and 5e-3 for text classification, in which the learning rate warm-up is 20% of the total steps. The $\lambda$ for balancing supervised and unsupervised learning is set to 1 in all our experiments. The supervised batch size is set to 4, and the unsupervised batch size is 32 for the summarization task (16 for the classification task) in our experiments.

## A.3 Evaluation Methodology

Extractive summarization quality is evaluated with ROUGE (Lin and Hovy, 2003). We report the full-length F1-based ROUGE-1, ROUGE-2, and ROUGE-L (R-1, R-2, and R-L), and these ROUGE scores are computed using `ROUGE-1.5.5.pl` script[7]. We report the accuracy (denoted as Acc) results in the text classification tasks.

## A.4 Baselines Details

For the text classification task, TinyBERT (Jiao et al., 2020) is a compressed model implemented by 6-layer or 4-layer $BERT_{BASE}$. For semi-supervised methods, we use the released code to train the UDA, which includes ready-made 12-layer $BERT_{BASE}$, 6-layer, or 4-layer TinyBERT. FLiText (Liu et al., 2021) is a lightweight and fast semi-supervised learning framework for the text classification task. FLiText consists of two training stages. It first

---

[6] https://github.com/google-research/bert
[7] https://github.com/andersjo/pyrouge

| DɪsCo ($S^{A6}$) | DɪsCo ($S^{B6}$) | Agnews w. 10 | | Yahoo!Answer w. 10 | | DBpedia w. 10 | |
|---|---|---|---|---|---|---|---|
| | | $S^{A6}$ | $S^{B6}$ | $S^{A6}$ | $S^{B6}$ | $S^{A6}$ | $S^{B6}$ |
| CKD:1,2,3,4,5,6 | CKD:7,8,9,10,11,12 | **80.71** | **81.05** | 56.61 | 55.08 | 98.04 | 98.12 |
| SKD:2,4,6,8,10,12 | CKD:1,2,3,4,5,6 | 74.45 | 74.38 | **59.10** | **57.62** | **98.57** | **98.50** |
| SKD:2,4,6,8,10,12 | SKD:1,3,4,5,7,9,11 | 79.67 | 80.13 | 56.50 | 56.73 | 97.84 | 97.79 |

Table 10: Text classification performance (Acc (%)) comparison with different combinations of model views in dual-student DɪsCo (6-layer TinyBERT as the students). SKD denotes the separated-layer knowledge distillation and CKD denotes connected-layer knowledge distillation.

Table 11: ROUGE performance of models using 10 or 1000 labelled CNN/DailyMail examples.

| Models | $L_m$ | $L_d$ | CNN/DailyMail | | |
|---|---|---|---|---|---|
| | | | R-1 | R-2 | R-L |
| CPSUM | 12 | 10 | 39.00 | **16.64** | 35.23 |
| UDASUM | 12 | 10 | 39.03 | 16.49 | 35.21 |
| UDASUM$_{TinyBERT^{A4}}$ | 4 | 10 | 38.67 | 16.62 | 35.23 |
| UDASUM$_{TinyBERT^{B4}}$ | 4 | 10 | 38.78 | 16.38 | 35.00 |
| DɪsCo ($S^{A4}$) | 4 | 10 | **39.20** | 16.51 | **35.34** |
| DɪsCo ($S^{B4}$) | 4 | 10 | 38.88 | 16.61 | 35.17 |
| CPSUM | 12 | 1000 | 40.42 | 17.62 | **36.59** |
| UDASUM | 12 | 1000 | 40.29 | 17.65 | 36.54 |
| UDASUM$_{TinyBERT^{A4}}$ | 4 | 1000 | 39.99 | 17.43 | 36.20 |
| UDASUM$_{TinyBERT^{B4}}$ | 4 | 1000 | 40.22 | 17.54 | 36.34 |
| DɪsCo ($S^{A4}$) | 4 | 1000 | **40.49** | **17.65** | 36.57 |
| DɪsCo ($S^{B4}$) | 4 | 1000 | 40.49 | 17.64 | 36.56 |

trains a large inspirer model (BERT) and then optimizes a target network (TextCNN).

Other SSL algorithms integrated with BERT are implemented in a unified semi-supervised learning benchmark (USB) (Wang et al., 2022a) for classification, including Mean Teacher (Tarvainen and Valpola, 2017), VAT (Miyato et al., 2019), FixMatch (Sohn et al., 2020), CRMatch (Fan et al., 2023), AdaMatch (Berthelot et al., 2022), and SimMatch (Zheng et al., 2022), all utilizing unlabeled data for invariant predictions. We report their text classification results in the USB benchmark testing. PCM (Xu et al., 2022) is a complex multi-submodule combination SSL model with three components, a K-way classifier, the class semantic representation, and a class-sentence matching classifier. MixText (Chen et al., 2020) is a regularization-based SSL model with an interpolation-based augmentation technique. Both PCM and MixText use a 12-layer BERT as the backbone model.

For extractive summarization, we extend TinyBERT and UDA for classifying every sentence, termed as UDASUM and TinyBERTSUM. Specifically, multiple [CLS] symbols are inserted in front of every sentence to represent each sentence and use their last hidden states to classify whether the sentence belongs to the summary. The SOTA semi-supervised extractive summarization model, CPSUM (Wang et al., 2022b), combines the noise-injected consistency training and the entropy-constrained pseudo labelling with the BERT$_{BASE}$ encoder. We also integrate the encoder of CPSUM with a slighter TinyBERT. It should be noted that the ORACLE system is an upper bound of the extractive summarization.

### A.5 Performance under Few-labels Settings

The form using differently labelled data in Table 2 indicates that there is a large performance gap between the 12-layer models and 4-layer models with only 10 labelled data due to the dramatic reduction in model size.

However, as shown in Table 11, in the extractive summarization tasks, DɪsCo works particularly well than the 12-layer models in the scenario of 100 labelled examples. The extractive summarization task is to classify every single sentence within a document, and the two views effectively encourage invariant prediction for unlabeled points' perturbations. DɪsCo achieves superior performance, as shown in Table 11, whether it uses only 10 or 1000 labelled data in extractive summarization. The superiority of DɪsCo with 4-layer BERT is more evident when processing 10 labelled extractive summarization, compared to CPSUM and UDASUM with 12-layer BERT. The results also indicate that our method can be suitable for extreme cases that suffer from severe data scarcity problems.

### A.6 More Different Model-view Analysis

We further investigate the performance of different model view combinations in dual-students DɪsCo. As described in section 2.2.1, the model view encoding has two forms: Separated-layer KD (SKD) and Connected-layer KD (CKD). Results of DɪsCo equipped with different model-view variants on the

| Models | Agnews | | | Yahoo!Answer | | | DBpeida | | |
|---|---|---|---|---|---|---|---|---|---|
| | 10 | 30 | 200 | 10 | 30 | 200 | 10 | 30 | 200 |
| DɪsCo ($S^{A4}$) | 76.90 | 85.39 | 87.82 | 51.48 | 62.36 | 68.10 | 94.02 | 98.13 | 98.56 |
| DɪsCo ($S^{B4}$) | 77.36 | 85.55 | 87.95 | 51.31 | 62.93 | 68.24 | **94.79** | 98.14 | 98.63 |
| Model Averaging Ensemble | **77.45** | **85.60** | **88.04** | **52.23** | **63.17** | **68.49** | 94.75 | **98.20** | **98.66** |

Table 12: Comparison of 4-layer DɪsCo with model averaging ensemble.

text classification tasks are summarized in Table 10. Although all three combinations of model views achieve improvement (compared to results in Table 2), the combinations of CKD and SKD for two students perform slightly better than other combinations. According to Sun et al. (2019), distilling across alternate $k$ layers in knowledge distillation captures more diverse representations, while distilling along connected $k$ layers tends to capture relatively homogeneous representations. By combining these two distinct strategies of model view encoding, DɪsCo acquires additional inductive bias for each student in the cohort, resulting in improved performance on downstream tasks.

### A.7 More Different Data-view Analysis

In Figure 3, we visualize the effect of DɪsCo (4-layer) integrating different data views encoding methods in the summarization task. We find that: DɪsCo integrating with the adversarial attack (AD) obtains superior performances, especially when data view is the adversarial attack in a SOFT FORM ($AD_A$, $AD_B$). DɪsCo with HARD FORM data views like ($AD_A$, $DO_B$) or ($DO_A$, $AD_B$) get sub-optimal effectiveness. This suggests that more advanced data augmentation methods pave the way for a more refined data view.

### A.8 Model Ensembling for Multiple Students

Model ensembling is an effective strategy, often yielding superior performance compared to individual models. As shown in Table 12, using simple model averaging for the 4-layer student model from Table 2 resulted in enhanced performance.

However, the core focus of our research is to ascertain the potential of a single model within our framework. Training requires two or more student models, but only one is essential for inference. Having multiple students during training ensures performance comparable to the teacher model, while selecting one student for inference upholds computational efficiency. Diving deeper into ensemble techniques to further amplify performance wasn't our primary objective.

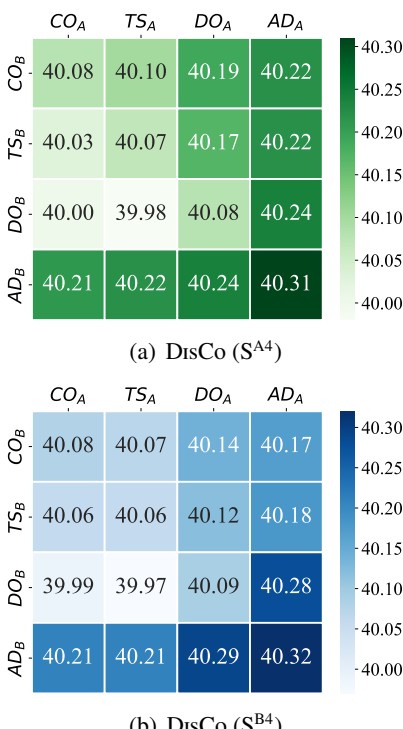

(a) DɪsCo ($S^{A4}$)

(b) DɪsCo ($S^{B4}$)

Figure 3: The performance visualization of the dual-student DɪsCo with data view using different combinations of data augmentation strategies. The row indicates the 1st data-augmentation-based data view encoding strategy, while the column indicates the 2nd data-augmentation-based data view encoding strategy. The results of dual-students DɪsCo with 4-layer TinyBERT being students are evaluated on the CNN/DailyMail with 100 labelled data.

### A.9 Selection of MSE or KL Loss

In our framework, we use the MSE loss to align the logits of the students. However, besides using MSE loss, employing Kullback-Leibler (KL) divergence to maintain consistency between the student predictions is also a widely chosen approach. We prefer the MSE loss in our framework because the student can learn better without suffering from the information loss that occurs when passing through logits to probability space (Ba and Caruana, 2014).

As shown in Table 13, the 4-layer DɪsCo with MSE loss performs better in the majority of cases. However, when labeled data is extremely limited

| Models | Agnews | | | Yahoo!Answer | | | DBpeida | | |
|---|---|---|---|---|---|---|---|---|---|
| | 10 | 30 | 200 | 10 | 30 | 200 | 10 | 30 | 200 |
| DisCo ($S^{A4}$) + MSE | 76.90 | 85.39 | 87.82 | 51.48 | 62.36 | 68.10 | 94.02 | 98.13 | 98.56 |
| DisCo ($S^{B4}$) + MSE | **77.36** | **85.55** | **87.95** | 51.31 | 62.93 | **68.24** | 94.79 | **98.14** | **98.63** |
| DisCo ($S^{A4}$) + KL | 76.46 | 83.76 | 87.20 | 52.90 | 61.81 | 67.17 | 95.63 | 97.72 | 98.38 |
| DisCo ($S^{B4}$) + KL | 77.31 | 83.94 | 87.17 | **53.68** | **63.21** | 67.68 | **96.14** | 97.83 | 98.41 |

Table 13: Comparison between MSE loss and KL divergence in 4-layer DisCo.

(e.g., 10 per class), KL divergence may surpass MSE in performance. This can be attributed to the noisy predictions produced by the student model, as its performance is not optimal because of the limited labeled data. KL divergence enforces label matching, thereby reducing issues resulting from corrupted knowledge transferred from another student model (Kim et al., 2021).

## A.10 Details in Loss Landscape Visualization

Our loss visualization approach adheres to the 'filter normalization' method (Li et al., 2018). For each setting, we select the top-performing student checkpoint based on its validation set results. Subsequently, we generate two random vectors and normalize them using parameters specific to each model. Ultimately, using the same training data and augmentation techniques, we plot the training loss landscape following the two normalized directions.