# OpenReview forum: "DisCo: Distilled Student Models Co-training for Semi-supervised Text Mining"
_EMNLP/2023/Conference — EMNLP 2023 Main_

### Official Review · Reviewer_VCv8 · 2023-07-31

**Soundness:** 3

**Excitement:**

4: Strong: This paper deepens the understanding of some phenomenon or lowers the barriers to an existing research direction.

**Paper Topic And Main Contributions:**

The authors propose a strategy for distilling and ensembling models in a semi-supervised setting, allowing for (a) faster inference in (b) smaller models, which (c) exhibit superior performance to comparable approaches. They evaluate their proposed modeling strategy on a set of benchmark data sets for classification and summarization and show superior performance to other small models as well as competitive performance to the full models.

**Questions For The Authors:**

A: L 330 ff: How did you come up with “the best model view and data view we found”? Was there any hyperparameter tuning? If yes: budget, search space, etc. (Appendix A2, which is referenced, does not provide any information)?

**Reasons To Accept:**

- The authors present a very convincing collection of ablations and thorough analyses of their models, extending it from two to multiple models, considering different options of data augmentations
- An effective SSL method allowing for leveraging unlabeled data is important in the current situation of ever-increasing amounts of unlabeled data


**Reasons To Reject:**

- The authors just state “1Code and data available at: [url redacted for blind review].” which is not sufficient imho; they could append it or make it available via an anonymous link. Further, I checked the pre-print of the paper (I still don’t know any of the authors, so no objections regarding anonymity), and not even there they provide a link to the code. So I am wondering what reasons they have to withhold access to their code.
- The explanation of “SOFT FORM” and “HARD FORM” is not really clear in the paper


**Reproducibility:**

1: Could not reproduce the results here no matter how hard they tried.

**Reviewer Confidence:**

3: Pretty sure, but there's a chance I missed something. Although I have a good feel for this area in general, I did not carefully check the paper's details, e.g., the math, experimental design, or novelty.

**Typos Grammar Style And Presentation Improvements:**

- I think there is a type in Figure 1 (left), where the outcome of the bottom one of the two blue students is denoted with “\bar y” instead of “\hat y”.
- l. 352: “Competiting” should be “Competing”?
- l. 360: Acronym CV not defined?
- l. 122: SOTA used w/o being defined; l. 366 “state-of-the-art” used

---

> ### Author Rebuttal · Authors · 2023-08-29
>
> Dear Reviewer VCv8,
>
> We would like to express our sincere gratitude for strongly supporting our contributions! Below, we provide a comprehensive response to each of your comments and concerns:
>
> > **C1: Concern about "the source code for future release"**
> - Thank you for your feedback. All our datasets and portions of our code related to the model architecture can be found in the Supplementary Materials. We will release the full code after the blind review, as providing links during this period is restricted.
>
> > **C2: Concern about "explanation of SOFT FORM and HARD FORM"**
> - The details related to "SOFT FORM" and "HARD FORM" are elaborated from lines 241 to 251.
>
> - Specifically, there are various data augmentation strategies, e.g., Adversarial Attack (AD), Cut Off (CO), and Dropout (DO). The "HARD FORM" combines **different strategies** for distinct data views, such as AD with DO. In contrast, "SOFT FORM" uses the **same strategy** across data views, like AD with AD. With the same strategy, different initializations can also produce varied data views.
>
> - Thank you for pointing it out. We will adjust the explanation in the final version to make it clearer.
>
> > **Q1 How did you come up with "the best model view and data view we found"?**
> - The specifics regarding selecting the best model view and data view are detailed in Appendix A5 and Appendix A6, respectively.
>
> - Regarding the best model view, we referred to established knowledge distillation methods [1,2,3] and **heuristically selected several plausible layer mapping strategies**. The selection of the best model view was then made based on the comparative performance of these strategies.
>
> - As for the best data view, we considered **pair-wise combinations of the four different data augmentation strategies discussed in the paper**. The optimal combination was determined based on the performance of these combinations.
>
> - Your concerned details will be introduced in the Appendix in our final paper.
>
> ---
>
> In conclusion, we once again thank you for your insightful critique and direction. Your feedback is instrumental in refining our paper.
>
> Best wishes,
>
> Authors of Paper5607
>
> **Reference:**
>
> [1] Xiaoqi Jiao, Yichun Yin, Lifeng Shang, Xin Jiang, Xiao Chen, Linlin Li, Fang Wang, and Qun Liu. TinyBERT: Distilling BERT for Natural Language Understanding. In Findings of the Association for Computational Linguistics: EMNLP 2020, 2020.
>
> [2] Wenhui Wang, Furu Wei, Li Dong, Hangbo Bao, Nan Yang, and Ming Zhou. MiniLM: Deep Self-Attention Distillation for Task-Agnostic Compression of Pre-Trained Transformers. In Advances in Neural Information Processing Systems, 2020.
>
> [3] Victor Sanh, Lysandre Debut, Julien Chaumond, and Thomas Wolf. DistilBERT, a distilled version of BERT: smaller, faster, cheaper and lighter. In CoRR, 2019.

---

### Official Review · Reviewer_vrX3 · 2023-08-03

**Soundness:** 3

**Excitement:**

3: Ambivalent: It has merits (e.g., it reports state-of-the-art results, the idea is nice), but there are key weaknesses (e.g., it describes incremental work), and it can significantly benefit from another round of revision. However, I won't object to accepting it if my co-reviewers champion it.

**Paper Topic And Main Contributions:**

This paper presents DisCo, a method to distill large language models into multiple small models with co-training for semi-supervised learning. DisCo tries to distill one large model into multiple small models by providing different distillation objectives, data transformations and distributions to build a committee of student models. During this process, DisCo also employs consistency training loss among students enable co-training among students. From the empirical results, the student models can outperform baseline models with similar size.

**Reasons To Accept:**

1. The authors provide comprehensive evaluation results, demonstrating the robustness of performance on the classification task.
2. The proposed training schema can bring insights for audience working on knowledge distillation.

**Reasons To Reject:**

1. Some technique choice in this paper is not suitable. For example, the authors use MSE to measure the difference between two probability distribution (the predictions on two models). On probability distribution difference, KL divergence or cross-entropy are more reasonable choice.
2. The variance between student models seems large according to table 2. This would raise a realistic change in model selection after distillation. Without accessing test set, we would not know which model performs better. The authors need to at least provide student model selected on validation set for each entry to give more realistic comparison.

**Reproducibility:**

3: Could reproduce the results with some difficulty. The settings of parameters are underspecified or subjectively determined; the training/evaluation data are not widely available.

**Reviewer Confidence:**

4: Quite sure. I tried to check the important points carefully. It's unlikely, though conceivable, that I missed something that should affect my ratings.

**Typos Grammar Style And Presentation Improvements:**

The multiple gigantic tables are a bit hard to follow and read. I suggest to subtract some entries and add the full table in appendix.

---

> ### Author Rebuttal · Authors · 2023-08-28
>
> Dear Reviewer vrX3,
>
> Thank you for your valuable feedback on our work. We appreciate your positive feedback and understand the concerns you have raised regarding some of the framework details, experiments, and presentation. Please find our detailed responses below:
>
> > **C1: Concern about "use MSE or KL loss"**
> - We apologize for the oversight. Equations 3, 4, and 7 contain typo errors. We actually **employ the MSE loss in the logits of the model**, not in the prediction. These will be corrected in the final version.
> - Our choice of loss function **aligns with previous SSL works**. In PI-Model and Temporal ensembling [1], the MSE loss was found to produce "better results than cross-entropy loss". Multiple SSL works, such as Mean Teacher [2] and Dual Student [3], follow this loss function.
>
> - The comparison between MSE and KL loss has been a topic of extensive discussion and research [4]. Given your concerns, we recognize the importance of this comparison and will endeavor to include it in the appendix of the final version.
>
> > **C2: Concern about 'model selection for inference'**
> - The subsequent table further illustrates the validation performance for the models presented in Table 2.  Given that all student model architectures are identical, **a model with better validation performance typically exhibits improved test results**.
>
> - Additionally, as shown in Tables 2, 4, and 6, regardless of the selection method—validation performance, heuristic methods, or even random—the student model consistently outperforms all baselines, ensuring robust final performance.
>
> - Choosing a single model for final inference based on validation performance is a common practice in methods involving multiple produced models, like mutual learning [5] and collaborative learning [6].
>
>  ***Table 1: Performance Comparison of DisCo on Validation and Test Sets***
>
> | Models                | dataset |           | Agnews    |           |           | Yahoo     |           |           | dbpedia   |           |
> | --------------------- | ------- | --------- | --------- | --------- | --------- | --------- | --------- | --------- | --------- | --------- |
> |                       |         | 10        | 30        | 200       | 10        | 30        | 200       | 10        | 30        | 200       |
> | DisCo ($\rm S^{A6}$ ) | Val     | 74.69     | 86.68     | 89.24     | 57.42     | 65.97     | 69.53     | 98.6      | 98.53     | 98.61     |
> |                       | Test    | 74.38     | 86.39     | 88.70     | 57.62     | 66.04     | 69.57     | 98.5      | 98.45     | 98.57     |
> | DisCo ($\rm S^{B6}$ ) | Val     | **77.40** | **87.14** | **89.45** | **58.48** | **66.47** | **69.87** | **98.63** | **98.60** | **98.77** |
> |                       | Test    | **77.45** | **86.93** | **88.82** | **59.10** | **66.58** | **69.75** | **98.57** | **98.61** | **98.73** |
> | DisCo ($\rm S^{A4}$ ) | Val     | 76.44     | 85.50     | **88.16** | 50.98     | 62.37     | 67.97     | 94.05     | **98.14** | **98.63** |
> |                       | Test    | 76.90     | 85.39     | 87.82     | **51.48** | 62.36     | 68.10     | 94.02     | 98.13     | 98.56     |
> | DisCo ($\rm S^{B4}$ ) | Val     | **77.41** | **85.68** | 88.11     | **51.25** | **63.24** | **68.42** | **94.78** | 98.11     | 98.62     |
> |                       | Test    | **77.36** | **85.55** | **87.95** | 51.31     | **62.93** | **68.24** | **94.79** | **98.14** | **98.63** |
> | DisCo ($\rm S^{A2}$ ) | Val     | 70.47     | 82.10     | 86.31     | 48.10     | 57.59     | 64.20     | 89.78     | 96.05     | 97.66     |
> |                       | Test    | 70.61     | 81.87     | 86.08     | 48.41     | 57.84     | 64.04     | **89.67** | 96.06     | 97.58     |
> | DisCo ($\rm S^{B2}$ ) | Val     | **74.30** | **82.24** | **86.74** | **50.95** | **58.36** | **65.69** | **89.80** | **96.25** | **97.82** |
> |                       | Test    | **75.05** | **82.16** | **86.38** | **51.05** | **58.83** | **65.63** | 89.55     | **96.14** | **97.70** |
>
> ---
>
> In addition, thank you for highlighting the issues with the tables and presentation. We will streamline and clarify them in the final version of the paper.
>
> Best wishes,
>
> Authors of Paper5607
>
> **Reference:**
>
> [1] Samuli Laine and Timo Aila. Temporal Ensembling for Semi-Supervised Learning. In 5th International Conference on Learning Representations, ICLR 2017, Toulon, France, April 24-26, 2017, Conference Track Proceedings, 2017.
>
> [2] Antti Tarvainen and Harri Valpola. Mean teachers are better role models: Weight-averaged consistency targets improve semi-supervised deep learning results. In 5th International Conference on Learning Representations, ICLR 2017, Toulon, France, April 24-26, 2017, Workshop Track Proceedings, 2017.
>
> [3] Zhanghan Ke, Daoye Wang, Qiong Yan, Jimmy S. J. Ren, and Rynson W. H. Lau. Dual Student: Breaking the Limits of the Teacher in Semi-Supervised Learning. In 2019 IEEE/CVF International Conference on Computer Vision, ICCV 2019, Seoul, Korea (South), October 27 - November 2, 2019, 2019.
>
> [4] Taehyeon Kim, Jaehoon Oh, Nakyil Kim, Sangwook Cho, and Se-Young Yun. Comparing Kullback-Leibler Divergence and Mean Squared Error Loss in Knowledge Distillation. In Proceedings of the Thirtieth International Joint Conference on Artificial Intelligence, IJCAI 2021, 2021.
>
> [5] Ying Zhang, Tao Xiang, Timothy M. Hospedales, and Huchuan Lu. Deep Mutual Learning. In Proceedings of the IEEE Conference on Computer Vision and Pattern Recognition (CVPR), 2018.
>
> [6] Siyuan Qiao, Wei Shen, Zhishuai Zhang, Bo Wang, and Alan Yuille. Deep Co-Training for Semi-Supervised Image Recognition. In Proceedings of the European Conference on Computer Vision (ECCV), 2018.

---

### Official Review · Reviewer_QmSR · 2023-08-05

**Soundness:** 4

**Excitement:**

4: Strong: This paper deepens the understanding of some phenomenon or lowers the barriers to an existing research direction.

**Missing References:**

Furlanello et al Born Again Neural Networks is probably relevant as it is example where student can improve over teacher.

**Paper Topic And Main Contributions:**

The paper presents a technique for knowledge distillation in the semi-supervised setting for BERT on multiple datasets. The approach jointly trains two students. By logit matching between the students, the setup enables effective use of the unlabeled data. The results are encouraging that this is a helpful KD recipe when there are only a small amount of labeled data. The paper includes extensive experiments, analysis, and details that should be helpful for replication and future work.

**Questions For The Authors:**

Questions about method.

* Why not use logits for the initial KD from the teacher step in Disco? This seemed to work pretty well for Distilbert, which retained 97% GLUE accuracy with 40% smaller model from bert-base. Albeit they used more supervised training data.

Questions about results.

* It seems somewhat awkward that Disco produces multiple students. Is the recommendation that we should only keep the better performing student based on some validation accuracy? Did you consider ensembling the students through model averaging or aggregating predictions or a MoE-like approach? That being said, utilizing multiple students at test time could lead to more resource requirements.

* How many random seeds for SingleStudent are used in Table 7? For Disco, it is almost as if there are two random seeds, so it seems it would be more fair if multiple seeds of SingleStudent were used as well. Also, did you consider ways for single student to leverage the unlabeled data?

* It would be helpful to have more details about the loss visualization. For example, how many data are used to draw the landscape? Is it using the same 10 labeled data from training?

* The loss landscape is visually appealing but I am not convinced it is that one is "much flatter" than the other.

* (low priority) More should be said about CNN/DailyMail when using 10/1000 in the main text. Otherwise perhaps remove mentions of the 10/1000 settings in the main text altogether.

**Reasons To Accept:**

Reason 1. The paper presents promising positive results for KD with BERT when there is small amount of labeled data but lots of unlabeled data. The approach is straightforward and depends on a combination of logit matching and layer matching. Many people would be interested in this approach as KD is often required to deploy NLP models. NOTE: Results are promising although perhaps not definitive that this is a new sota method for the semi-supervised setting.

Reason 2. The extensive analysis, results, and ablations will definitely be useful for researchers using this and similar KD work. Using more than two students is particularly interesting.

Reason 3. More specifically, the comparison between model and data-views will help researchers understand the important conceptual components when designing KD pipelines.

**Reasons To Reject:**

Reason 1. Although the two student approach is convenient, it remains unclear if two students are necessary. For example, finetuning the initial teacher then providing pseudo labels for the student seems like an obvious baseline --- I think this baseline was only included for Agnews where it does outperform Disco. I have more question about single student in the question section.

Reason 2. The Disco approach has some awkwardness, since the output is two student models --- if we do not have much validation data, then it might be hard to pick the better model among the two. Perhaps more statistics could be reported about performance or a protocol for combining models. This would make the findings more actionable.

Reason 3. It's unclear if the tested datasets are "low resource". Perhaps datasets that are more likely represent real world semi-supervised data would be more informative.

Reason 4. Sometimes the text is a bit strong wrt claims. For instance, I am not sure table 6 is showing anything "unequivocally" about the Disco performance.

**Reproducibility:**

5: Could easily reproduce the results.

**Reviewer Confidence:**

4: Quite sure. I tried to check the important points carefully. It's unlikely, though conceivable, that I missed something that should affect my ratings.

**Typos Grammar Style And Presentation Improvements:**

Style

* (very low priority) I found the blue bold to be distracting and confusing at times. It is a personal choice, but I think the paper is easier to read with the blue numbers removed. Especially since sometimes blue bold text is not the best.
* (very low priority) I could also do without the average in Table 1. It's confusing to average 10/30/200 together.

---

> ### Author Rebuttal · Authors · 2023-08-28
>
> Dear Reviewer QmSR,
>
> We would like to thank you for all the meticulous and constructive feedback and positive comments. We have elaborated on our responses as follows:
>
> > **Q1: Why not use logits for the initial KD from the teacher step in Disco?**
>
> - As outlined in Section 2.2, our choice for the teacher step KD was to adopt a **task-agnostic KD**, following the approach used by TinyBERT [1].
>
>     The primary goal of this task-agnostic KD is to offer a good and general initialization for the compact student model tailored for downstream tasks. This eliminates the need to repeatedly fine-tune the large teacher model and then distil it to the student across different datasets, ensuring computational efficiency.
>
> - Given that the prediction head in the task-agnostic KD is irrelevant to the downstream tasks, distilling logits from the teacher step was deemed non-essential for our methodology.
>
> > **Q2: Concern about "Disco produces multiple students"**
>
> - Training requires two or more student models, but **only one is essential for inference**. Having multiple students during training ensures performance comparable to the teacher model, while selecting one student for inference maintains computational efficiency.
>
> - Selecting a single model for final inference based on validation performance is commonly adopted in methods involving multiple produced models, such as mutual learning [2] and collaborative learning [3].
>
> - Model ensembling is indeed an effective strategy (as shown in the subsequent table). However, our primary focus is to determine **the potential of a singular model** within our framework (a single student for inference already obtains noticeable performances). Exploring ensemble techniques that can seamlessly integrate various perspectives of the student model presents a promising direction for future work.
>
> - Moreover, as demonstrated in Table 2 and Table 4 and Table 6, regardless of whether we pick the student model based on validation performance or heuristic methods, its results consistently outperform baselines, ensuring robust final performance.
>
> ***Table 1: Comparison of 4-layer DisCo with Model Averaging Ensemble***
> | Models                   |           | Agnews    |           |           | Yahoo     |           |           | dbpedia   |           |
> |--------------------------|-----------|-----------|-----------|-----------|-----------|-----------|-----------|-----------|-----------|
> |                          | 10        | 30        | 200       | 10        | 30        | 200       | 10        | 30        | 200       |
> | DisCo ($\rm S^{A4}$ )    | 76.90     | 85.39     | 87.82     | 51.48     | 62.36     | 68.10     | 94.02     | 98.13     | 98.56     |
> | DisCo ($\rm S^{B4}$ )    | 77.36     | 85.55     | 87.95     | 51.31     | 62.93     | 68.24     | **94.79** | 98.14     | 98.63     |
> | Model Averaging Ensemble | **77.45** | **85.60** | **88.04** | **52.23** | **63.17** | **68.49** | 94.75     | **98.20** | **98.66** |
>
> > **Q3: How many random seeds for SingleStudent**
>
> - For SingleStudent, we **consistently used three random seeds**, ensuring uniformity in our experimental process. The model and dataset settings from the 6-layer DisCo in Table 7 are the same as those in Table 2.
>
> - Incorporating unsupervised data directly during SingleStudent fine-tuning presents challenges. SSL techniques like consistency constraints or mix-up are typically employed to utilize unsupervised data.
>
> - (can be discussed) Furthermore, the dual-student DisCo, without the model view, can essentially be perceived as co-training SingleStudent alongside its duplicate, making optimal use of unlabeled data. Performance comparisons are shown in Table 8.
>
> > **Q4, 5: Concern about 'loss visualization'**
>
> - More details about loss visualization will be added to the appendix in the final version. We follow the "filter normalization" method [4]. We select the top-performing student checkpoint and use **the same training data and augmentation techniques** for plotting.
>
> - (can be discussed) Reviewers' perspectives are expected to be rigorous. As for the notion of a flatter loss landscape, we are open to engaging in a thorough discussion.
>
>     Employing the "filter normalization" method ensures our results remain unaffected by the scale of the model parameters. Therefore, achieving a distinctly "flatter" result can prove challenging given identical model architecture and training data.
>
>     In light of this, we acknowledge that our statement might be a "strong claim". We will modify the claim in the final version.
>
> ---
>
> In addition, thank you for highlighting the concerns regarding the presentation and the omission of certain references. We appreciate your feedback and will make the necessary revisions (e.g., "strong claims") in the final version to address these issues.
>
> Best wishes,
>
> Authors of Paper5607
>
> **Reference:**
>
> [1] Xiaoqi Jiao, Yichun Yin, Lifeng Shang, Xin Jiang, Xiao Chen, Linlin Li, Fang Wang, and Qun Liu. TinyBERT: Distilling BERT for Natural Language Understanding. In Findings of the Association for Computational Linguistics: EMNLP 2020, 2020.
>
> [2] Ying Zhang, Tao Xiang, Timothy M. Hospedales, and Huchuan Lu. Deep Mutual Learning. In Proceedings of the IEEE Conference on Computer Vision and Pattern Recognition (CVPR), 2018.
>
> [3] Siyuan Qiao, Wei Shen, Zhishuai Zhang, Bo Wang, and Alan Yuille. Deep Co-Training for Semi-Supervised Image Recognition. In Proceedings of the European Conference on Computer Vision (ECCV), 2018.
>
> [4] Hao Li, Zheng Xu, Gavin Taylor, Christoph Studer, and Tom Goldstein. Visualizing the Loss Landscape of Neural Nets. In Advances in Neural Information Processing Systems, 2018.

---

### Meta-Review · Area_Chair_WV8G · 2023-09-18

**Recommendation:** 4

**Metareview:**

The paper introduces DisCo, a semi-supervised learning framework designed for fine-tuning a set of smaller models derived from a pre-trained language model via knowledge distillation. The primary innovation is in the co-training technique, which encourages knowledge sharing amongst the student models based on varied views produced by differing distillation strategies and input augmentations.

Strengths:
The paper's innovative approach with KD in scenarios with limited labeled data was seen as promising, particularly its results and methodology (QmSR). The results are found to be promising, especially on the classification task, and the proposed training schema is found to provide new insights for those in the knowledge distillation domain (vrX3). The introduction of an effective SSL method is deemed crucial (Reviewer VCv8).

Weaknesses:
Concerns were raised regarding the two-student approach, with questions surrounding its necessity and the complications it introduces in model selection due to the potential variance between student models (QmSR, vrX3). The datasets used for testing were scrutinized for their authenticity in representing "low resource" scenarios, and some claims, such as those in table 6, were seen as possibly overstated (QmSR). Some design choices like MSE were questioned and variance of some of the results seemed too large (vrX3).

---

### Decision · Program_Chairs · 2023-10-07

**Decision:**

Accept-Main

**Comment:**

The paper introduces DisCo, a semi-supervised learning framework designed for fine-tuning a set of smaller models derived from a pre-trained language model via knowledge distillation. The primary innovation is in the co-training technique, which encourages knowledge sharing amongst the student models based on varied views produced by differing distillation strategies and input augmentations.

Strengths:
The paper's innovative approach with KD in scenarios with limited labeled data was seen as promising, particularly its results and methodology (QmSR). The results are found to be promising, especially on the classification task, and the proposed training schema is found to provide new insights for those in the knowledge distillation domain (vrX3). The introduction of an effective SSL method is deemed crucial (Reviewer VCv8).

Weaknesses:
Concerns were raised regarding the two-student approach, with questions surrounding its necessity and the complications it introduces in model selection due to the potential variance between student models (QmSR, vrX3). The datasets used for testing were scrutinized for their authenticity in representing "low resource" scenarios, and some claims, such as those in table 6, were seen as possibly overstated (QmSR). Some design choices like MSE were questioned and variance of some of the results seemed too large (vrX3).